# Paraoxonase 2 (PON2) Deficiency Reproduces Lipid Alterations of Diabetic and Inflammatory Glomerular Disease and Affects TRPC6 Signaling

**DOI:** 10.3390/cells11223625

**Published:** 2022-11-16

**Authors:** Henning Hagmann, Naghmeh Hassanzadeh Khayyat, Cem Oezel, Antonios Papadakis, Alexander Kuczkowski, Thomas Benzing, Erich Gulbins, Stuart Dryer, Paul T. Brinkkoetter

**Affiliations:** 1Department II of Internal Medicine, University of Cologne, Faculty of Medicine and University Hospital Cologne, 50937 Cologne, Germany; 2Center for Molecular Medicine Cologne, University of Cologne, Faculty of Medicine, University Hospital Cologne, 50931 Cologne, Germany; 3Department of Biology and Biochemistry, University of Houston, Houston, TX 77204, USA; 4Institute for Genetics, Faculty of Mathematics and Natural Sciences, University of Cologne, 50931 Cologne, Germany; 5Cologne Cluster of Excellence on Cellular Stress Responses in Ageing-Associated Diseases (CECAD) and Systems Biology of Ageing Cologne (Sybacol), 50931 Cologne, Germany; 6Department of Molecular Biology, University Hospital Essen, University of Duisburg-Essen, 45147 Essen, Germany; 7Department of Biomedical Sciences, Tilman J. Fertitta Family College of Medicine, University of Houston, Houston, TX 77204, USA

**Keywords:** podocyte, calcium signaling, oxidative stress, lipid peroxidation

## Abstract

Diabetes and inflammatory diseases are associated with an altered cellular lipid composition due to lipid peroxidation. The pathogenic potential of these lipid alterations in glomerular kidney diseases remains largely obscure as suitable cell culture and animal models are lacking. In glomerular disease, a loss of terminally differentiated glomerular epithelial cells called podocytes refers to irreversible damage. Podocytes are characterized by a complex ramified cellular architecture and highly active transmembrane signaling. Alterations in lipid composition in states of disease have been described in podocytes but the pathophysiologic mechanisms mediating podocyte damage are unclear. In this study, we employ a genetic deletion of the anti-oxidative, lipid-modifying paraoxonase 2 enzyme (PON2) as a model to study altered cellular lipid composition and its effects on cellular signaling in glomerular disease. PON2 deficiency reproduces features of an altered lipid composition of glomerular disease, characterized by an increase in ceramides and cholesterol. PON2 knockout mice are more susceptible to glomerular damage in models of aggravated oxidative stress such as adriamycin-induced nephropathy. Voltage clamp experiments in cultured podocytes reveal a largely increased TRPC6 conductance after a membrane stretch in PON2 deficiency. Correspondingly, a concomitant knockout of TRPC6 and PON2 partially rescues the aggravated glomerular phenotype of a PON2 knockout in the adriamycin model. This study establishes PON2 deficiency as a model to investigate the pathophysiologic mechanisms of podocyte dysfunction related to alterations in the lipid composition, as seen in diabetic and inflammatory glomerular disease. Expanding the knowledge on these routes and options of intervention could lead to novel treatment strategies for glomerular disease.

## 1. Introduction

Diseases affecting renal glomeruli comprise the most common causes of end stage renal failure, resulting in the need for renal replacement therapy [1]. Glomerular epithelial cells, referred to as podocytes, are essential for the integrity and function of the glomerular filter. Podocytes are highly specialized terminally differentiated multipolar cells located on the outer surface of the glomerular capillary tuft. A number of fine processes emanate from the podocyte cell body and terminate in foot processes that adhere to the external surface of the glomerular capillary. Foot processes interdigitate with each other by means of specialized adhesion proteins such as nephrin, thereby forming a specialized ribbon-like junction known as the slit diaphragm. The slit diaphragm represents the final barrier to the water and solute flow across the glomerular filter. Podocyte dysfunction leads to the retraction of primary and secondary foot processes, a process known as effacement, resulting in the loss of plasma proteins, especially albumin, into the urine (proteinuria/albuminuria) [2]. In animal models, this can also be induced by drugs such as adriamycin or puromycin aminonucleoside (PAN) [3,4]. A number of different transmembrane signaling processes in the vicinity of the slit diaphragm regulate the foot process cytoskeleton and maintain the delicate ultrastructure of the glomerular filter [5,6,7].

Transmembrane signaling events mediated by receptors and channels have long been known to be affected by the composition of the surrounding plasma membrane. Transient receptor potential-6 (TRPC6) is a multimodal cation channel and has been implicated in inherited and acquired glomerular diseases. The activation of TRPC6 has been most extensively studied in the context of signaling through G-protein coupled receptors via the activation of phospholipase C (PLC) [8,9,10,11]. The localized generation of plasma membrane diacyl-glycerol (DAG) and NOX2-mediated reactive oxygen species (ROS) increase the activity and expression of TRPC6 on the cell surface [12,13]. However, it should be noted that endogenous TRPC6 can also be activated in response to the membrane stretch [14,15]. TRPC6 is expressed in both the cell body and in the foot process slit diaphragm domains of podocytes [16,17,18,19]. Certain mutations in the TRPC6 gene, which most frequently result in a gain of channel function, cause podocyte diseases, including a subset of the cases of familial focal segmental glomerulosclerosis (FSGS) (reviewed in [8]). Increases in the glomerular TRPC6 expression have also been observed in primary forms of FSGS, in membranous nephropathy and diabetic nephropathy, as well as in animal disease models such as the PAN nephrosis model in rats [13,20,21]. 

During states of inflammation and in diabetes, free oxygen and nitrogen radicals cause membrane damage by a series of chain reactions that lead to the peroxidation of lipids and an altered cellular lipid metabolism. The association of glomerular diseases with an altered cellular lipid metabolism and modifications of cellular lipids is increasingly recognized. The accumulation of cholesterol and sphingolipids, especially ceramide, has been detected in diabetic nephropathy [22,23,24,25]. A similar process occurs in podocytes in several disease states (reviewed in [26]). Whether the alteration of the lipid content is a cause or effect of glomerular dysfunction remains unclear, pathophysiological mechanisms driven by sphingolipid accumulation are not well understood, and suitable animal models are lacking [27].

Cells typically express an armamentarium of enzymes that can terminate lipid peroxidation at various cellular levels, including a family of enzymes known as paraoxonases. Among these, paraoxonase-2 (PON2) is a ubiquitously expressed type-2 transmembrane protein that translocates to the plasma membrane in response to increases in intracellular Ca^2+^ [28]. The enzymatic domain of PON2 is ectofacial [28,29,30], catalyzes hydrolysis of arylesters and lactones, and reduces localized oxidative processes [31]. Oxidized membrane phospholipids have been shown to accumulate with ceramide and cholesterol in macrophages and neurons [32,33,34]. Polymorphisms of PON2 are associated with hypercholesterolemia, coronary artery disease, and cerebrovascular disease [35,36,37]. Interestingly, the phenotypes of PON2 polymorphisms seem to be fueled by oxidative stress. Accordingly, the PON2 variants A184G and S311C are associated with diabetic nephropathy in patients with type II diabetes and obesity [38].

The paraoxonase family of enzymes is evolutionarily highly conserved [39]. The nematode ortholog of PON2 is named MEC-6. It localizes to the mechanosensory complex of *C. elegans* where it interacts with the mechanosensitive degenerin Na^+^ channel MEC-4/-10 to regulate its channel activity [40]. The nematode mechanosensory complex shares certain structural homologues with the mammalian slit diaphragm and relies on a distinct lipid environment to maintain its function [41]. Besides MEC-6, the stomatin family protein MEC-2, which is an orthologue to the mammalian slit diaphragm protein podocin, orchestrates the lipid environment of MEC-4/-10 [19]. In mammals, podocin was shown to affect nephrin and TRCP6 signaling at the slit diaphragm [42,43]. A regulatory effect of PON2 on the TRPC6 channel function seems plausible. 

In the present study, we introduce a model of lipid peroxidation of the cellular membranes induced by abrogating the expression of PON2 in cultured podocytes and in mouse models of glomerular disease. We hypothesized that podocyte damage can be driven by an altered lipid composition through actions on ion channels such as TRPC6. We show that a PON2 deficiency results in large increases in the activation of TRPC6 induced by a membrane stretch, which is accompanied by a prolonged time to return to the channel resting state. In addition, we show that a PON2 deficiency aggravates the glomerular phenotype in the adriamycin nephropathy model in mice, and that this occurs through a TRPC6-dependent mechanism.

## 2. Material and Methods

### 2.1. Immunohistochemistry

Paraffin-embedded human kidney tissue samples were cut to into 4 µm sections and stained with a PON2-specific polyclonal antibody (Proteintech, Chicago, IL, USA). Briefly, the sections were deparaffinized in Xylene (VWR, Darmstadt, Germany). After rehydration in graded ethanol and the blocking of endogenous peroxidases with 3% hydrogen peroxide (Fischer, Saarbrücken, Germany), the sections were incubated overnight at 4 °C with a primary antibody diluted in 1% bovine serum albumin (BSA)/phosphate-buffered saline (PBS). The sections were washed repeatedly in PBS before an incubation with biotinylated anti-rabbit secondary antibody (Jackson Immuno-Research, West Grove, PA, USA) diluted in 1% BSA/PBS for 1 h at room temperature. The ABCkit (Vector, Burlingame, CA, USA) was used for the signal amplification, and 3,3-diaminobenzamidine (Sigma-Aldrich, St. Louis, MO, USA) was used as a chromogen. The slides were counterstained with hematoxylin (Sigma-Aldrich), dehydrated, and covered with Histomount (National Diagnostics, Atlanta, GA, USA). Samples of human kidney tissue were obtained from spare material after clinical histologic analysis. All the procedures were approved by the Internal Review Board of the Medical Faculty, University of Cologne (DRKS00024517) and conducted in accordance with the Declaration of Helsinki. The patients gave informed consent.

### 2.2. Cell Culture and shRNA-Induced Knockdown of PON2

Murine podocytes (kind gift from S. Shankland, University of Washington) were maintained as previously described [44]. Podocytes were seeded on plastic dishes coated with collagen I. Podocyte cell lines express the SV40 large T temperature-sensitive antigen. The cells were cultured and maintained at 33 °C (termed “undifferentiated” cells). Transferring the cells to 37 °C induces a cell cycle arrest and differentiation. The cells were left at 37 °C for 10 days and are referred to as “differentiated” podocytes at this stage. The RNA interference experiments were done as described previously [45]. In brief, short hairpin RNAs (shRNAs) against the murine PON2 3′-untranslated region were designed based on the prediction of publicly available prediction programs (RNAi Designer; Invitrogen, Waltham, MA, USA) and multiple shRNA sequences tested in luciferase assays as previously described [46]. Stable cell lines were generated by a transduction with lentivirus generated from constructs in the pLenti6.3 backbone. 

The following sequences were used:

mPON2 hp#1: 5′_TGCTGTTAAATTCCCTCAGAATTGGCGTTTTGGCCACTGACTGACGCCAATTCAGGGAATTTAA_3′.

mPON2 hp#2: 5′_TGCTGATTACTAGGTCAGTACTGATGGTTTTGGCCACTGACTGACCATCAGTAGACCTAGTAAT_3′.

The re-expression of murine PON2 in PON2-deficient murine podocytes was achieved by a retroviral gene transfer using the mPON2 coding sequence in the pBabe vector.

### 2.3. LC MS/MS Analysis of Phospholipids Contained in Podocyte Membranes

The podocyte cell membranes were purified as previously described [28]. Briefly, PON2-proficient and -deficient cells were lysed in a hypotonic buffer (10 mM HEPES, 1.5 mM MgCl2, 10 mM KCl, 0.5 mM DTT, and 0.05% Nonidet P-40, pH 7.9), homogenized in a Dounce glass/glass homogenizer, and subjected to a sequential centrifugation. Nuclei and mitochondria were segregated from the sample by a centrifugation at 15 × 10^3^× *g*, and membrane fractions [endoplasmatic reticulum (ER) and plasma membrane] were collected as the pellet after a final ultracentrifugation step at 10^5^× *g* in a Beckman TLA 55 rotor (Beckman Coulter, Fullerton, CA, USA). The membrane pellets were resuspended in water and the lipids were extracted using chloroform/methanol 2:1 (*v*/*v*) (Sigma-Aldrich) and internal standards (67 pmol PC 17:0–14:1, 81 pmol PE 17:0–14:1, 63 pmol PI 17:0–14:1, 68 pmol PS 17:0–14:1, 77 pmol PG 17:0–14:1, 82 pmol PA 17:0–14:1 (Avanti Polar Lipids)). The sample was sonicated and vigorously vortexed and the phase separation was induced by a centrifugation (800× *g* for 5 min at 4 °C). After repetitive extractions, the organic phases were combined and dried under a stream of nitrogen. The dried lipid extracts were analyzed by LC-ESI-MS/MS as previously described.

### 2.4. Acid Sphingomyelinase Activity

The samples were shock frozen, thawed and immediately lysed in 200 µL of 250 mM sodium acetate (pH 5.0) and 1% NP40 for 5 min. The tissue samples were then sonicated three times for 10 secs each using a tip sonicator. The lysates were diluted to 0.2% NP40 in 250 mM of sodium acetate (pH 5.0) and a 60 µL aliquot was added to 0.05 µCi [14C]sphingomyelin (52 mCi/mmol; Perkin Elmer, #NEC 663010UC) per sample. The substrate [14C]sphingomyelin was dried for 10 min in a SpeedVac, resuspended in 250 mM of sodium acetate (pH 5.0) and 0.1% NP40, and sonicated for 10 min in a bath sonicator before it was added to the samples. The samples were incubated for 30 min at 37 °C with shaking at 300 rpm. The reaction was stopped by organic extraction in 4 volumes of CHCl3:CH3OH (2:1, *v*/*v*), the samples were vortexed, centrifuged, and an aliquot of the upper aqueous phase was scintillation-counted to determine the release of [14C]phosphorylcholine from [14C]sphingomyelin.

### 2.5. Measurement of Ceramide

The samples were lysed in 200 µL of H_2_O using 3 rounds of tip sonication and the lysates were extracted in CHCl3:CH3OH:1N HCl (100:100:1, *v*/*v*/*v*). The phases were separated by 5 min at a 14,000× *g* rpm centrifugation of the samples. The lower phase was collected, dried, and resuspended in 20 µL of a detergent solution (7.5% [*w*/*v*] n-octyl glucopyranoside, 5 mM of cardiolipin in 1 mM of diethylenetriaminepentaacetic acid [DTPA]). The samples were bath-sonicated for 10 min to obtain micelles and were immediately used for the kinase reaction. The kinase reaction was initiated by adding 70 µL of a reaction mixture containing 10 µL of diacylglycerol (DAG) kinase (GE Healthcare Europe, Munich, Germany), 0.1 M of imidazole/HCl (pH 6.6), 0.2 mM of DTPA (pH 6.6), 70 mM of NaCl, 17 mM of MgCl2, 1.4 mM of ethylene glycol tetraacetic acid, 2 mM of dithiothreitol, 1 µM of adenosine triphosphate (ATP), and 5 µCi [32P] γATP (6000 Ci/mmol; Hartmann Radiochemicals, Braunschweig, Germany). The kinase reaction was performed for 60 min at room temperature with 300 rpm shaking. The samples were then organically extracted in 1 mL of CHCl3:CH3OH:1N HCl (100:100:1, *v*/*v*/*v*), 170 µL of buffered saline solution (135 mM of NaCl, 1.5 mM of CaCl2, 0.5 mM of MgCl2, 5.6 mM of glucose, and 10 mM of HEPES [pH 7.2]) and 30 µL of a 100 mM EDTA solution were added and the phases were separated. The lower phase was collected, dried in a SpeedVac, separated on Silica G60 thin-layer chromatography (TLC) plates with chloroform/acetone/methanol/acetic acid/H_2_O (50:20:15:10:5, *v*/*v*/*v*/*v*/*v*), and developed with a Fuji phosphoimager (Fujifilm, Tokyo, Japan). The ceramide amounts were determined by a comparison with a standard curve using C16/18 to C22/C24 ceramides as substrates.

### 2.6. Expressional Analysis

The isolation of glomeruli and the preparation of a single cell suspension was performed as previously described [47] on 11-week-old PON2-deficient mice in the CD1 background and wildtype littermates. Highly parallel single-cell transcriptome profiling was performed by Drop-seq as published previously [48]. The raw sequencing reads from 10× Genomics 3-prime sequencing were processed with Cell Ranger [49] to generate the gene-cell barcode UMI matrices. The cells expressing less than 1000 UMIs and more than 5000 UMIs were discarded. Additionally, only cells with more than 300 genes and less than 10% mitochondrial transcripts were retained. The final dataset contained 34295 cells with a median of 1233 genes and 1982 unique molecular identifiers per cell. The gene expression analysis and cell type identification were performed using Seurat V4.0 [50]. For the normalization, we used the SCTransform function from Seurat and the resulting values were used for the downstream dimensional reduction (PCA). The first thirty components of the PCA were used for the construction of a shared nearest neighbors graph and the cell clustering was done via the Louvain algorithm (functions FindNeighbors and FindCluster from the R package Seurat V4.2.0). The identification of cluster markers was performed with a Wilcoxon test on the log-normalized counts. These markers were used for the manual cell-type identification of podocyte cells. The expression of genes of interest was visualized specifically in podocytes.

### 2.7. Immunoprecipitation and Immunoblot

HEK 293T cells were transiently transfected with V5-tagged and FLAG-tagged constructs as indicated. After an incubation for 48 h, the cells were lysed in lysis buffer (10 mM of Tris pH 8.0, 150 mM of NaCl, 50 mM of NaF, and 2% SDS in ddH_2_O) (all from Sigma-Aldrich) in the presence of protease inhibitors (Roche, Basel, Switzerland). The lysates were homogenized by sonication with a digital sonifier (Branson Ultrasonics, Danbury, CT, USA). The TRPC6 protein was precipitated from the samples by using a specific antibody directed against the V5 or FLAG-epitope tag on proteinA-coated Sepharose beads. The samples were separated on SDS-PAGE, then protein blotted on PVDF membranes, followed by an immunoblot analysis using specific antibodies detecting the V5 epitope tag (Bio-Rad, Hercules, CA, USA) and the FLAG-epitope tag (Sigma-Aldrich). The visualization of immunoblots was achieved with chemiluminescence.

### 2.8. Electrophysiology

Whole-cell recordings were made at room temperature (22 °C) from immortalized podocytes, as described in detail previously [14,15,51] using fire-polished borosilicate glass microelectrodes (4–6 MΩ) and an Axopatch 1D amplifier (Molecular Devices, Foster City, CA, USA). The bath was perfused at a constant flow rate (0.3 mL/min) and TRPC6 channels were activated by a bath application of a hypotonic stretch solution as described previously [14]. The macroscopic currents were monitored during 2.5 sec voltage ramps from −80 mV to +80 mV from a holding potential of −40 mV, and currents at +80 mV were quantified for the statistical analysis. We have previously shown that the stretch-evoked cation currents are eliminated by the TRPC6 knockdown [14] and by the application of agents such as SAR-7334 that selectively inhibit TRPC6 [8]. 

### 2.9. Animals Models

The PON2 knockout mice were a kind gift from Sven Horke, University of Mainz, Germany [52]. The TRPC6 knockout mice were obtain from Alexander Dietrich, University of Munich (LMU), Germany (MTA in place) [53]. The mice were housed according to the standardized specific pathogen-free conditions in the University of Cologne’s animal facility. All animal experiments were performed in accordance with the guidelines provided by the LANUV NRW (Landesamt für Natur, Umwelt und Verbraucherschutz Nordrhein-Westfalen/State Agency for Nature, Environment and Consumer Protection North Rhine-Westphalia, approval numbers: 2013.A375, 2019.A067). Adriamycin nephropathy was induced in 11-week-old male mice. Anesthesia/analgesia was performed with isoflurane/buprenorphine. After a single intra-venous injection of adriamycin (15 mg/kg body weight), the animals were followed for up to seven weeks and the urinary albumin excretion was analyzed longitudinally on days 0, 14, and 49 as the albumin/creatinine ratio from the spot urine samples. The quantification of albuminuria was performed using commercial kits (mouse albumin ELISA kit; Bethyl Labs, Montgomery, TX and urine creatinine assay; Cayman Chemical, Ann Arbor, MI, USA). To quantify the serum creatinine concentration, blood samples were centrifuged at 3000× *g* rpm and 4 °C for 10 min and 100 mL of serum was taken from the supernatant and analyzed for creatinine levels in the central laboratory of the University Hospital Cologne, Germany. The renal tissue was embedded in an OCT compound (Sakura, Torrance, CA, USA) or snap frozen in liquid nitrogen and stored at minus 80 °C or fixed in 10% neutral buffered formalin for immunostaining.

## 3. Results

### 3.1. Expression of PON2 Is Induced in Diabetic and Inflammatory Glomerular Diseases

PON2 protects tissues from lipid peroxidation and it is induced in states of oxidative stress [54,55,56]. We stained for PON2 on the kidney samples from patients with proteinuric kidney diseases such as minimal change nephropathy (MCN), early-stage diabetic nephropathy, and systemic inflammatory diseases such as systemic lupus erythematodes (SLE) and ANCA-associated vasculitis (AAV). In the healthy kidney, the PON2 expression is primarily localized in tubular cells (Figure 1) [57]. In samples from patients with early diabetic kidney disease (DKD), the PON2 reactivity is increased in glomeruli and is especially notable in podocytes (Figure 1 DKD). In lupus nephritis class V (lupus membranous nephropathy) and in ANCA-associated vasculitis samples, PON2 staining is also stronger in podocytes (Figure 1. SLE + ANCA). In contrast, no enhanced PON2 reactivity was detected in the glomeruli of patients suffering from nephropathy due to arterial hypertension or minimal change nephropathy (Figure 1 HTN + MCN). The quantification of the PON2 positive cells per glomerular area shows significantly more PON2 expressing cells in DKD, ANCA, and SLE compared to the other conditions (Appendix A). 

### 3.2. PON2 Deficiency Affects the Lipid Composition of Podocytes

Several models of the altered cellular lipid metabolism have been examined in podocytes [58,59]. In the present study, we established a PON2-deficient podocyte cell line by a lentiviral gene transfer of hairpin siRNA (validation of PON2 knockdown shown in Appendix A), and we analyzed the cellular lipid composition by mass spectrometry. In concordance with the previously reported data, a PON2-deficiency is associated with an accumulation of cholesterol and a reduction in DAG, whereas the triglyceride content is not altered (Appendix A) [60,61]. In addition, the profiling of sphingolipids and phospholipids by LC MS/MS analysis of the murine podocyte cell lysates reveals an accumulation of ceramide and sphingomyelin in PON2-deficient cells (Figure 2). In contrast, phosphatidylcholine, phosphatidylethanolamine, and phosphatidylinositol are reduced in PON2-deficient cells whereas phosphatidylglycerol is increased compared to the control cell line. No significant differences in phosphatidylserine or phosphatidic acid were detected. 

### 3.3. PON2 Protects Mice from Glomerular Damage in Adriamycin-Induced Nephropathy

The PON2 knockout mouse does not develop a kidney phenotype in the absence of other manipulations. Specifically, we did not detect albuminuria or histologic changes even in the advanced age of 72 weeks (data not shown). Therefore, to promote oxidative stress on cellular membranes, we employed the model of adriamycin-induced nephropathy to PON2 knockout mice as well as to wildtype and heterozygous littermates. Notably, in the CALINCA dataset, PON2 was induced while most other anti-oxidative enzymes were reduced in adriamycin-treated mice compared to the vehicle controls [62]. In addition, the ceramide metabolism was induced after the adriamycin treatment. In the present study, the lipid MS/MS analysis of the kidney cortex of adriamycin-treated PON2 knockout and wildtype animals detected an increased ceramide content in PON2-deficient mice (Figure 3A). The accumulation of C16/C18 ceramide is even more pronounced than the accumulation of the C22/24 ceramide (Figure 3B). The single nucleus RNA sequencing of the isolated glomeruli of PON2-deficient mice and wildtype littermates revealed a comparably abundant nephrin (NPHS1) expression, while the PON2 expression is expectedly not detectable in podocytes of PON2-deficient mice (Figure 3C). The expression levels of ceramide generating ceramide synthase 6 (Cers6) and acid sphingomyelinase (SMPD1/2) are comparable in PON2-deficient podocytes and the wildtype. The ceramidases Acer2 and Asah1 are only slightly reduced in PON2-deficient mice. In contrast, the quantitative functional analysis of the sphingomyelinase activity shows an induced enzymatic activity in the PON2 knockout kidneys (Figure 3D). Sphingomyelinase hydrolyses sphingomyelin to ceramide and phosphorylcholine. 

On day 14 after the adriamycin treatment, the wildtype and heterozygous mice developed borderline nephrotic range albuminuria. Those animals fully recovered by day 49. By contrast, PON2 knockout animals exhibit severe albuminuria on day 14 which does not recover by day 49 (Figure 4A). The animals were euthanized on day 49 after the administration of adriamycin for the subsequent analyses. The serum creatinine levels in PON2 knockout mice are substantially greater than those measured in the wildtype and heterozygous animals (*p* < 0.05 ko vs. wt) (Figure 4B). However, no significant difference is noted in the serum urea levels (Figure 4C). The histological changes in the kidneys from adriamycin-treated wildtype, heterozygous, and PON2-knockout mice are subtle (Figure 4D), with knockout animals showing adhesions at the Bowman’s capsule and foam cell degeneration of the podocytes. However, glomerulosclerosis is not detected in any group. Electron microscopy imaging reveals the foot process effacement in PON2 ko mice while the PON2 wildtype mice show a regular foot process alignment (Figure 4E).

In the animal model of puromycin aminonucleoside nephrosis, oxidative stress and the increased activation of TRPC6 are well established [63,64]. In addition, the inactivation of TRPC6 has been shown to be protective in a rat model of PAN nephrosis [51]. Changes in TRPC6 during adriamycin nephropathy are less well studied, although the mechanism of damage is most likely related. In voltage clamp experiments on cultured mouse podocytes after an exposure to 0.5 µg/mL of adriamycin for 24 h, the activation of TRPC6 is induced by a superfusion with hypotonic solution as compared to vehicle treated murine podocytes (Appendix A).

### 3.4. PON2 Deficiency Augments TRPC6 Channel Activity

The gating of the TRPC channels varies depending on the local environment of the membrane lipids, and this is considered to be the canonical mechanism of their activation [65]. For this reason, we hypothesized that PON2 can impact the functional properties of the TRPC6 channels in podocytes. Immunoprecipitation experiments after the co-expression of FLAG-tagged PON2 and V5-tagged slit diaphragm proteins in HEK 293T cells revealed a strong interaction of PON2 and TRPC6. Interactions are also found between PON2 and podocin and between PON2 and nephrin (Figure 5A). No interaction is detected between PON2 and CD2AP. 

The functional effects of PON2 on TRPC6 were examined in whole-cell voltage clamp recordings from PON2-proficient and PON2-deficient differentiated mouse podocytes. We have previously shown that a hypotonic stretch causes the activation of the TRPC6 channels in podocytes [8,14]. In PON2-deficient cells, we observed that a hypotonic membrane stretch evokes very large cationic currents (Figure 5B,C). We have previously noted that the stretch-induced activation of TRPC6 in wildtype podocytes is rapidly reversible, with currents returning to the baseline in 5 min or less after the bath is switched back to a normotonic solution [14]. By contrast, the TRPC6 currents are not only larger, but they remain active for a much longer time after a return to normotonic salines in PON2-deficient podocytes (Figure 5C). Thus, the cation currents in those cells always remain markedly elevated by 15 min after a return to normotonic salines, and a complete recovery is usually required around 40 min. As with PON2-proficient cells, the stretch-evoked currents in PON2-deficient cells are blocked by the pan-TRP inhibitor SKF-96365, and also by 100 μM of La^3+^, which blocks TRPC6 but which enhances the activation of TRPC5, another cation channel expressed in podocytes [66] (Appendix A). Cationic currents are also recorded in cells treated with the diacylglycerol analog (OAG 100 µM) for 30 min. As reported previously [67], those currents were greater than those recorded from the cells maintained in a normal media for the same period of time (Appendix A). We note here that this effect of OAG is significantly greater in PON2-deficient cells. Thus, the two-way ANOVA, in which the independent variables were (1) an OAG treatment or not and (2) a normal or reduced expression of PON2, indicated that OAG causes a significant increase in the current in both types of cells. However, there is a significant interaction effect between the effects of the expression of PON2 and the effects of OAG, indicating that the effect of OAG is proportionately greater in PON2-deficient cells (Appendix A). 

### 3.5. TRPC6 Knockout Partially Rescues the Proteinuria Phenotype of PON2 Knockout in Adriamycin-Induced Nephropathy

Transferring these observations back into the mouse model, we crossed the PON2 knockout mouse to a conventional TRPC6 knockout mouse. PON2 knockout was validated by qPCR (Appendix A). TRPC6 knockout was validated by immunofluorescence staining of kidney sections employing TRPC6-specific antibody (Appendix A). Again, no baseline phenotype is observed (data not shown). To reduce the number of animals required in the adriamycin-induced disease model, we compared PON2/TRPC6 double knockout mice (PON2/TRPC6 dko) with PON2-deficient littermates that carried TRPC6 as a heterozygous allele (PON2ko/TRPC6het). As previously, an adriamycin challenge was applied by a single intravenous injection and proteinuria was quantified at the baseline and at days 14 and 49 after the treatment with adriamycin. Nephrotic range proteinuria was observed in PON2ko/TRPC6het mice, while PON2/TRPC6 dko mice showed significantly reduced proteinuria at day 14 (Figure 6). At day 49, both PON2/TRPC6 dko mice as well as the PON2ko/TRPC6het exhibited nephrotic range proteinuria and had to be sacrificed due to the weight loss criteria. The serum creatinine levels at day 49 were increased in PON2ko/TRPC6het, while they were normal in PON2/TRPC6 dko (mean 0.34 mg/dL vs. 0.1 mg/dL) (Figure 6B).

## 4. Discussion

There is a growing body of literature showing that changes in lipid dynamics contribute to the pathogenesis of podocyte diseases. For example, a ceramide accumulation has been described in glomerular disease [59,68,69,70]. An impaired cholesterol efflux and an increased cholesterol uptake in glomerular cells has been documented in diabetic nephropathy-comprising studies of human samples as well as genetic and sporadic animal models of glomerular disease [71,72,73,74]. PON2 is relevant in this context because it can counteract a cholesterol accumulation in several cell types [60,61]. In the present study, we have shown that a podocyte PON2 abundance is increased in human diabetic and inflammatory glomerular disease. In addition, we found that a PON2 deficiency drives a cholesterol accumulation and ceramide levels in cultured podocytes and in the kidney cortex of mice after the induction of oxidative stress. Recent studies show that alterations in the lipid metabolism, specifically the accumulation of ceramides, affects the function of podocyte. The podocyte-specific knockout of lysosomal ceramidase resulted in a ceramide accumulation and nephrotic range proteinuria, whereas the simultaneous knockout of acid sphingomyelinase rescued the phenotype [59]. Other studies have shown that the inhibition of ceramide synthesis by the knockout of acid sphingomyelinase or pharmacologic inhibition was protective in animal models of high fat-induced and diabetic glomerular disease [68,69]. In the present study, enhanced acid sphingomyelinase (ASM) activity was detected as a source of a ceramide accumulation during a PON2 deficiency.

Interestingly, we observed that a PON2 deficiency alone is not sufficient to produce a glomerular phenotype. However, after the aggravation of lipid peroxidation by an adriamycin administration, PON2-deficient mice exhibit sustained nephrotic range proteinuria. The accompanying histologic changes were subtle, although it bears noting that those are known to vary quite substantially depending on the dose of adriamycin applied and the mouse’s genetic background [75]. In the present study, we titrated the dose of adriamycin to yield transient low nephrotic range albuminuria in wildtype mice while still allowing for a reasonable tolerability and survival in PON2-deficient mice. That dose window in our mouse population was very narrow.

In addition to lipid dynamics, there is now an extensive body of evidence depicting that sustained increases in the TRPC6 activity can contribute to the progression of multiple forms of glomerular disease [11,16,17,20,76]. TRPC6 is a multimodal cation channel that becomes active in response to chemical stimuli such as diacylglycerol, but which can also become active in response to physical stimuli that cause a stretch or local deformation of the surrounding plasma membrane [14,77]. Recent studies suggest that both modes of TRPC6 activation may be important in podocytes in vivo [78]. For this reason, extensive drug development efforts have focused on TRPC6 as a therapeutic target for glomerular diseases and also for renal fibrosis [79,80,81]. Unfortunately, many of the current inhibitors of TRPC6 have been limited by either a lack of specificity or a poor bioavailability [82,83]. A better understanding of the pathophysiologic mechanisms leading to an excessive TRPC6 activation in podocytes could lead to novel therapeutic strategies. 

In the present study, we examined a connection between cellular lipid dynamics and the activity of TRPC6. The canonical pathway for the activation of TRPC6 is through G-protein-coupled receptor (GPCR) pathways, for example, the activation of the TRPC6 channel via angiotensin 2 (Ang II) or ATP through the activation of phospholipase C (PLC) [8,9,10,11]. Indeed, it has long been known that TRPC6 can be activated by diacylglycerol analogs even in excised membrane patches [84]. More recent studies have shown that in podocytes DAG and its analog OAG, they induce the formation of NOX2 complexes at the plasma membrane, which allows for the generation of ROS in the direct vicinity of TRPC6-containing protein complexes at the slit diaphragm [12]. These studies emphasize the role of TRPC6 as redox-sensitive channels, as its activation by Ang II and ATP is blocked by a NOX2 inhibition or knockdown and by ROS-scavengers [9], whereas it can be enhanced by the application of H_2_O_2_ [12]. 

Independent of GPCR or local ROS signaling, TRPC6 can become active in response to the membrane stretch [14,15]. In an experimental setup using cultured podocytes, an exposure to hypotonic (70%) solution leads to cell swelling and increases the stretch of the plasma membrane, resulting in cation currents that are abolished by highly selective TRPC6 inhibitors and which are markedly reduced following the TRPC6 knockdown [8]. These TRPC6-mediated currents persist in the presence of ROS quenchers and are normally reversible within 5 min of a replacement of the external solution with an isotonic medium [14]. In the present study, we observed that stretch-induced currents via TRPC6 were extensively altered in PON2-deficient podocytes. Specifically, peak currents were several-fold larger and the recovery of the channel activity after a previous stretch activation was prolonged to around 40 min in PON2-deficient podocytes. Some of these effects resemble those of a podocin knockdown, which also results in marked increases in the amplitude of TRPC6 responses to the membrane stretch [14]. However, a podocin knockdown does not affect the time-course of the recovery from the stretch-induced channel’s activation. In addition, a podocin knockdown reduces the activation of podocyte TRPC6 channels by OAG, whereas the responses to OAG are actually increased in PON2-deficient podocytes. It is well established that the plasma membrane lipid composition is a crucial regulator of the function of TRPC6, including effects on both trafficking and gating [19,43,77]. A limitation of the present study is that we were not able to analyze the lipid interactors of TRPC6 directly. The purification of the TRPC6 protein was not possible without applying harsh detergent concentrations, most likely due to the firm integration of the TRPC6 protein in membrane domains, and the detergent solubilized interacting lipids and the LC MS/MS analysis failed. 

The present study suggests that TRPC6 contributes to the deleterious effects of a PON2 knockout in the adriamycin-induced nephropathy mouse model, at least at early stages of the disease. This conclusion is based on the marked reduction in proteinuria seen after the simultaneous knockout of TRPC6 and PON2. This conclusion is limited by the fact that the knockouts of PON2 and TRPC6 were global and not podocyte-specific. In addition, adriamycin is known to affect several organ systems. The transient nature of the protection afforded by the TRPC6 knockout suggests that a PON2-deficiency affects other cellular processes that progress over time, but presently we do not know what these are.

There is still a lack of therapeutic options for most glomerular diseases. In recent years, alterations of the lipid metabolism in glomerular cells have gained attention. Currently, the first clinical phase II trials targeting the altered lipid metabolism in glomerular diseases are being conducted. The lipid-modifying small molecule R3R01 affects the cellular lipid content and is tested in patients with steroid resistant FSGS and uncontrolled proteinuria in Alport syndrome (NCT05267262). However, the pathophysiology referring podocyte disease in states of the altered lipid metabolism are not yet understood. Our data suggest a pathophysiologic mechanism involving increased TRPC6 activity in response to the altered lipid content and substantiate the significance of research directed to the lipid metabolism in glomerular disease to open novel therapeutic avenues. 

In summary, PON2-deficiency mimics altered lipid dynamics, which are usually seen in diabetic and inflammatory glomerular disease. The present study delineates a role for lipid peroxidation, resulting in podocyte damage through mechanisms that are at least in part related to TRPC6. The unopposed oxidation of membrane lipids in a PON2 deficiency results in an increased TRPC6 activation in terms of both the amplitude and time-course. Drugs that affect the activation of TRPC6 through effects on the local membrane environment could represent a therapeutic strategy for certain glomerular diseases.

## Figures and Tables

**Figure 1 cells-11-03625-f001:**
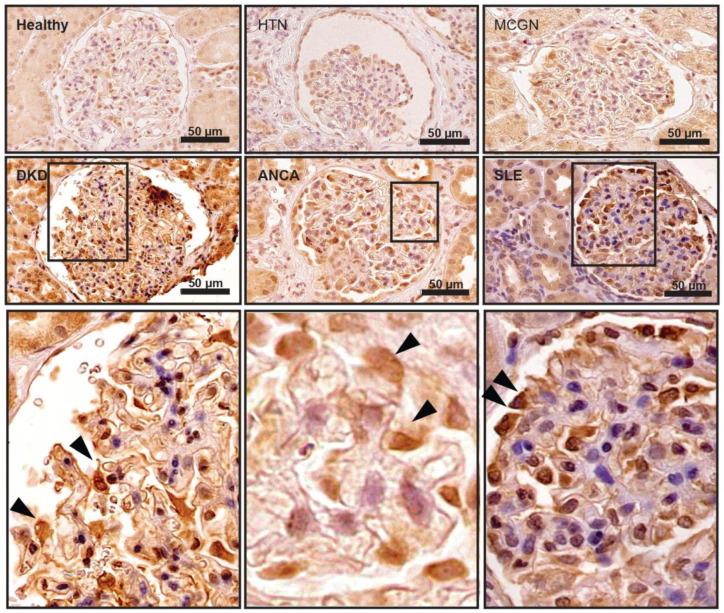
PON2 expression is induced in diabetic and inflammatory human glomerular kidney disease. Human kidney biopsy samples stained with PON2 specific antibody. PON2 reactivity is primarily localized in tubular cells in healthy kidney as well as in hypertensive (HTN) and minimal change (MCGN) glomerular disease while glomeruli show faint staining. PON2 staining is pronounced in glomeruli in diabetic kidney disease (DKD), ANCA vasculitis (ANCA), and lupus nephritis (SLE). Specifically, PON2 expression is predominant in podocytes (arrow heads) localizing to the outer aspect of the glomerular tuft.

**Figure 2 cells-11-03625-f002:**
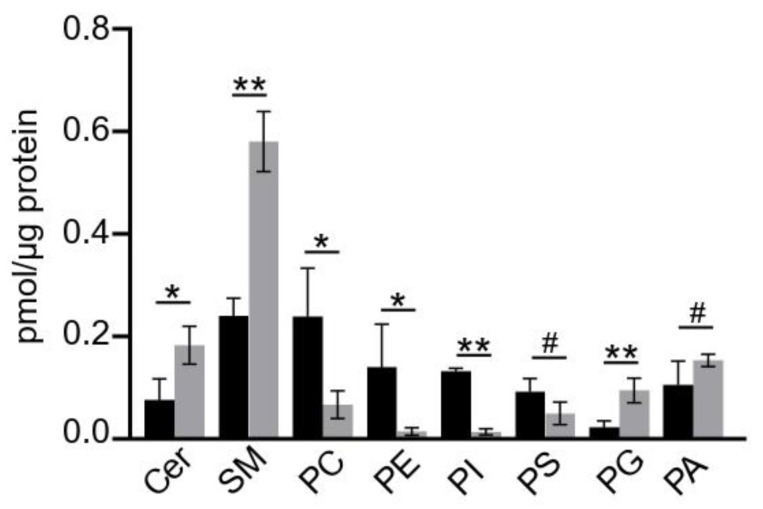
PON2 deficiency alters lipid composition of cultured podocytes. Cellular sphingo- and phospholipids content of PON2-deficient and -proficient murine podocytes was assessed by LC MS/MS. PON2-deficient podocytes (grey bars) contain more ceramide and sphingomyelin compared to the control (black bars) (ceramide: mean 0.183 pmol/µg protein; SD 0.037 vs. mean 0.076 pmol/µg protein; SD 0.042; * *p* < 0.05; sphingomyelin: mean 0.581 pmol/µg protein; SD 0.059 vs. mean 0.240 pmol/µg protein; SD 0.035; ** *p* < 0.01). Content of phosphatidylcholine, phosphatidylethanolamine, and phosphatidylinositol are markedly reduced in PON2-deficient podocytes (* *p* < 0.05; ** *p* < 0.01; # not significant).

**Figure 3 cells-11-03625-f003:**
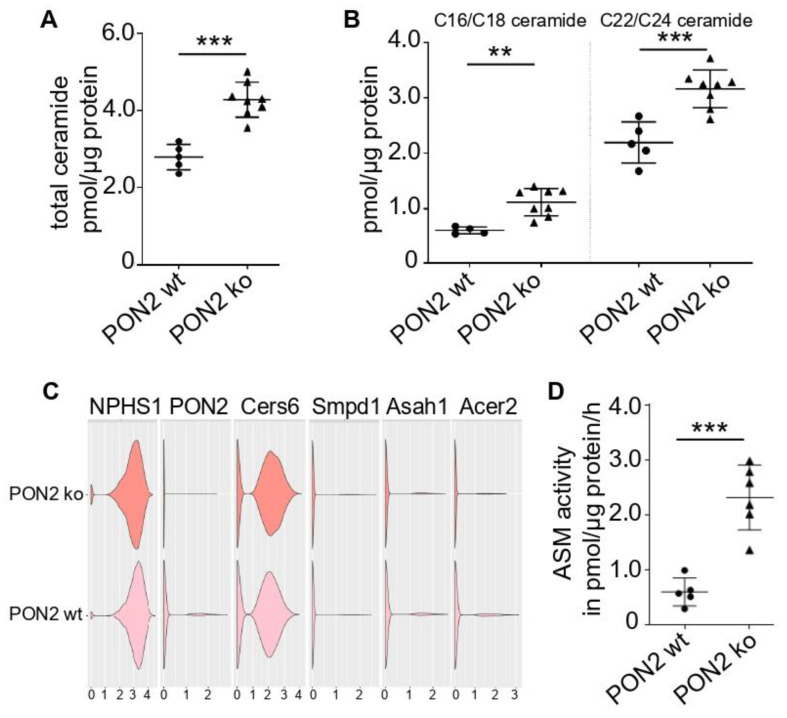
PON2 deficiency alters lipid composition in adriamycin-induced nephropathy kidney cortex samples of adriamycin-treated mice were subjected to LC MS/MS to quantify ceramide content and subclasses. (**A**) Total ceramide content is significantly increased in PON2 ko (triangles) as compared to wildtype mice (dots) (*n* = 8 and 5, respectively). (**B**) Both C16/C18 ceramide and C22/C24 ceramide are increased in PON2 ko compared to control. (**C**) Expression profiling of PON2-proficient and -deficient podocytes investigating RNA levels of NPHS1, ceramide generating enzymes Cers6 and SMPD1/2, and ceramidase enzymes Asah1 and Acer2. (**D**) Acid sphingomyelinase (ASM) activity was assessed in cortex samples of PON2 ko and wildtype mice. ASM activity is largely increased in PON2 ko vs. PON2 wt (*n* = 6 and 5, respectively) (** *p* < 0.01; *** *p* < 0.001).

**Figure 4 cells-11-03625-f004:**
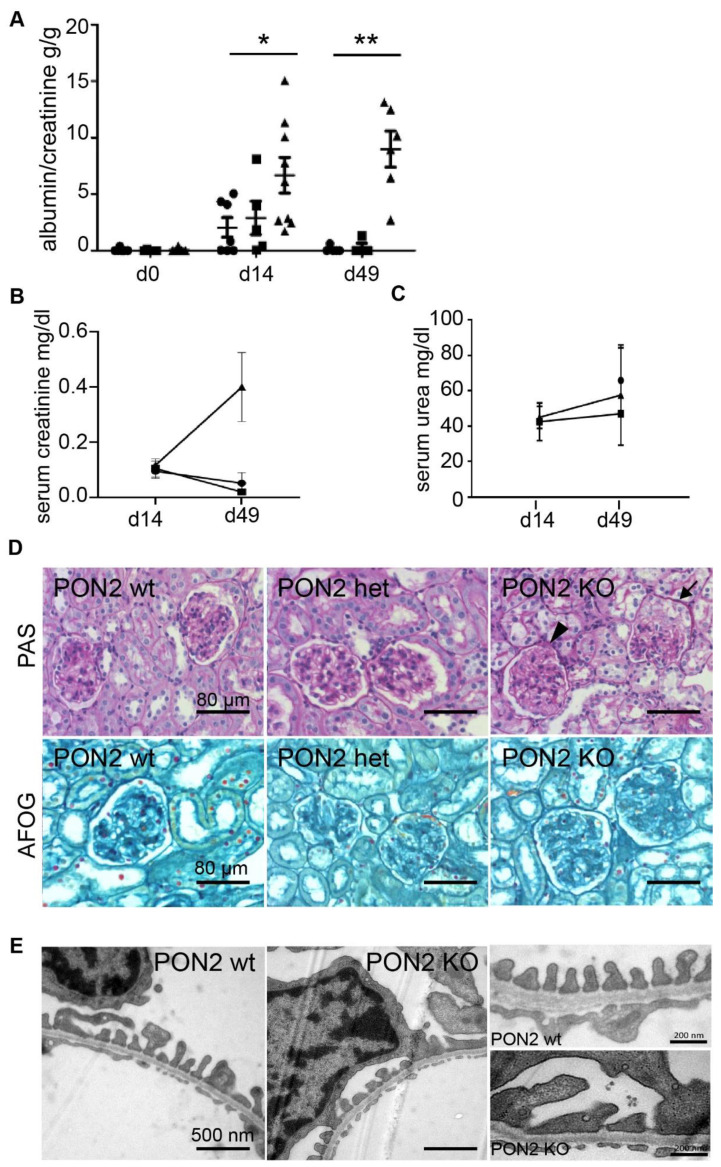
PON2 deficiency aggravates the glomerular phenotype of adriamycin-induced nephropathy. Quantification of albuminuria in the AIN model on day 0, day 14, and day 49 after adriamycin induction (**A**). No albuminuria is detected at baseline. Wildtype (circles, n7) and PON2 heterozygous mice (squares, n6) develop subnephrotic range albuminuria on day 14 and recover to baseline on day 49. PON2 ko mice (triangles, d14 n9, and d49 n6) show nephrotic albuminuria on day 14 which aggravates until day 49. Two-way ANOVA with *p* < 0.001 for genotype, days after treatment and interaction. (* *p* < 0.05; ** *p* < 0.01) (**B**,**C**) Serum creatinine and serum urea on day 14 and 49. At day 49 serum creatinine is significantly increased in PON2 ko mice compared to wildtype and PON2 heterozygous mice (* *p* < 0.05; *n* = 5 animals each group). No significant differences for serum urea levels are detected. (**D**) Representative images of histologic analysis of kidney sections on day 49 do not reveal glomerular sclerosis in any of the samples. In PAS stainings, glomerular adhesion to the Bowman’s capsule (arrowhead) and foam cell degeneration of podocytes (arrow) are present in samples of PON2 ko animals. (**E**) Representative electron microscopy images of kidney samples on day 49 show foot process effacement on PON2 ko mice whereas wildtype mice present regular foot process architecture.

**Figure 5 cells-11-03625-f005:**
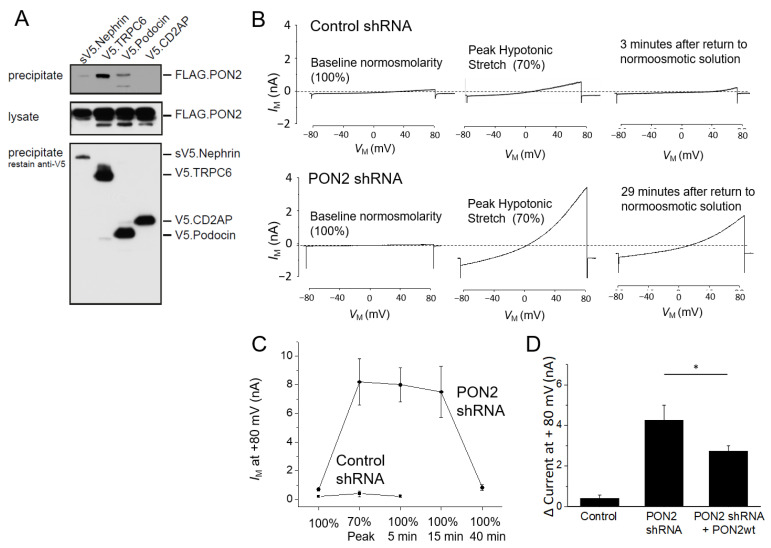
TRPC6 conductance is regulated by PON2. (**A**) Immunoprecipitation of FLAG-tagged PON2 and V5-tagged slit diaphragm proteins reveals interaction of PON2 and TRPC6, Podocin, and Nephrin and no interaction with CD2AP. (**B**) Representative recordings of whole cell voltage clamp experiments using PON2-proficient and -deficient murine podocytes. TRPC6 currents in response to membrane stretch after hypotonic stimulation (70%) are largely increased in PON2-deficient podocytes. (**C**) Currents at +80 mV during whole cell recordings including medium change from 100% tonicity to 70% tonicity and back. In PON2-deficient podocytes currents stay active for 40 min, while control cells recover after 5 min. (**D**) Enhanced TRPC6 conductance of PON2-deficient podocytes is reduced by re-expression of PON2 wildtype (* *p* < 0.05) (control mean 0.39 nA, SD 0.16; PON2 shRNA mean 4.26 nA, SD 0.73; PON2 shRNA + PON2 wt 2.73 nA, SD 0.26).

**Figure 6 cells-11-03625-f006:**
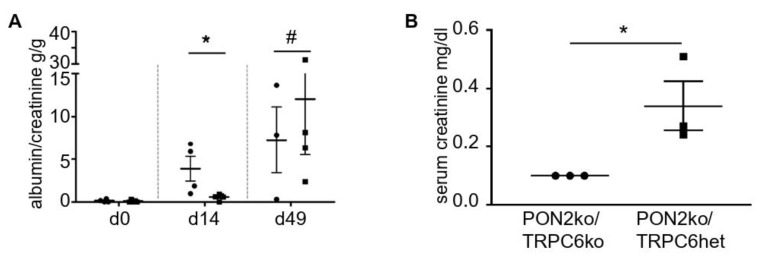
Aggravation of the adriamycin-induced nephropathy phenotype in PON2 knockout is mediated by TRPC6. (**A**) Quantitative analysis of albuminuria in PON2/TRPC6 double knockout (squares) and PON2 ko/TRPC6 heterozygous (circles) mice. While PON2ko/TRPC6het mice develop albuminuria on day 14, PON2ko/TRPC6ko mice are protected (*n* = 4 and 5, respectively; * *p* < 0.05, # not significant). In the further course, mice of both genotypes show progressive nephrotic proteinuria. (**B**) Quantification of serum creatinine on day 49 show increased concentrations in PON2ko/TRPC6het mice as compared to PON2ko/TRPC6ko (*n* = 3; * *p* < 0.05).

## Data Availability

Data are available upon request to corresponding authors.

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
