# Peer review of "Paraoxonase 2 (PON2) Deficiency Reproduces Lipid Alterations of Diabetic and Inflammatory Glomerular Disease and Affects TRPC6 Signaling"

_cells, 2022, doi:10.3390/cells11223625_

Round 1

Reviewer 1 Report

This is a novel study examining the effects of paraoxonase 2 (PON2) enzyme deficiency on podocyte lipid composition, TRPC6 activity in podocytes and the response to podocyte injury in an vivo in a model of nephrotic syndrome. The overall hypothesis that PON2 deficiency alters membrane composition which then affects TRPC6 activity is interesting but the manuscript is significantly lacking in mechanistic detail and some of the data provided do not provide a coherent story.

The  authors initially show that PON2 expression is increased in human glomeruli from patients with diabetes and autoimmune kidney disease but then go on to examine PON2 deficiency in mouse podocytes in vitro and in vivo. They show that silencing PON2 in cultured podocytes alters membrane lipid composition and that TRPC6 conductance is significantly increased after hypertonic-induced membrane stretch in PON2-deficient podocytes compared to controls but don't provide any mechanistic explanation for this.

They also find that PON2 KO in mice leads to a sustained increase in albuminuria after adriamycin injury and that this increase can be partially reversed by TRPC6 KO but the mechanism behind these findings is not clear as there are minimal histologic alterations by light microscopy in the glomeruli of PON2 KO mice and no EM has been performed on these mice.

Specific concerns include the following:

1) If Figure 1 is retained in the manuscript, quantitation of PON2 expression from multiple glomeruli from healthy patients or those with diabetes, ANCA vasculitis or HTN/MCD should be done.

2) Data should be shown confirming lentiviral mediated knockdown of PON2 in the PON2 siRNA podocyte line.

3) Data should be shown demonstrating that PON2 and TRPC6 are absent in the double KO line.

4) The number of mice for the albuminuria, serum creatinine and BUN plots in Figure 4 should be shown.

5) What does the interaction between PON2 - an enzyme that regulates lipids - and nephrin, a slit diaphragm protein or TRPC6 a membrane bound channel mean?

6) Is calcium signaling altered in PON2 deficient podocytes?

Author Response

We thank the reviewers for their thoughtful comments. We provide a point-by-point response with our reply in italics.

Best regards,

for all authors-

Henning Hagmann

Reviewer #1:

This is a novel study examining the effects of paraoxonase 2 (PON2) enzyme deficiency on podocyte lipid composition, TRPC6 activity in podocytes and the response to podocyte injury in an vivo in a model of nephrotic syndrome. The overall hypothesis that PON2 deficiency alters membrane composition which then affects TRPC6 activity is interesting but the manuscript is significantly lacking in mechanistic detail and some of the data provided do not provide a coherent story.

The authors initially show that PON2 expression is increased in human glomeruli from patients with diabetes and autoimmune kidney disease but then go on to examine PON2 deficiency in mouse podocytes in vitro and in vivo. They show that silencing PON2 in cultured podocytes alters membrane lipid composition and that TRPC6 conductance is significantly increased after hypertonic-induced membrane stretch in PON2-deficient podocytes compared to controls but don't provide any mechanistic explanation for this.

The reviewer’s critique is well taken. We would like to point out, that we submitted two manuscripts for back-to-back publication that build on one another. Mechanistic investigations focusing on membrane biophysics are presented in the manuscript entitled “Capsazepine (CPZ) inhibits TRPC6 conductance and is protective in adriamycin-induced nephropathy and diabetic glomerulopathy.”

They also find that PON2 KO in mice leads to a sustained increase in albuminuria after adriamycin injury and that this increase can be partially reversed by TRPC6 KO but the mechanism behind these findings is not clear as there are minimal histologic alterations by light microscopy in the glomeruli of PON2 KO mice and no EM has been performed on these mice.

We thank the reviewer for this important comment. We have now included electron microscopy data of PON2 wt and PON2 ko animals showing aggravated foot process effacement in PON2 ko. We have amended the text and figure legends accordingly.

Specific concerns include the following:

1) If Figure 1 is retained in the manuscript, quantitation of PON2 expression from multiple glomeruli from healthy patients or those with diabetes, ANCA vasculitis or HTN/MCD should be done.

The reviewer brings up helpful criticism. We have included quantification of PON2-positive cells per glomerular area in a novel Suppl. Fig. 1.

2) Data should be shown confirming lentiviral mediated knockdown of PON2 in the PON2 siRNA podocyte line.

We kindly ask to refer to former Suppl. Fig. 1 B, now novel Suppl.Fig. 2B which shows a western blot of control shRNA and PON2-directed shRNA expressing cells stained for PON2 confirming lentiviral mediated knockdown of PON2.

3) Data should be shown demonstrating that PON2 and TRPC6 are absent in the double KO line.

We thank the reviewer for this insightful comment. We have included qPCR data for PON2 to validate the knockout of PON2 and staining for TRPC6 on kidney sections to substantiate TRPC6 knockdown. This data is shown in novel Suppl. Fig. 6. Both mice are conventional knockout mice that have been analyzed frequently by different groups in the past.

4) The number of mice for the albuminuria, serum creatinine and BUN plots in Figure 4 should be shown.

We have now provided these numbers in the figure legend.

5) What does the interaction between PON2 - an enzyme that regulates lipids - and nephrin, a slit diaphragm protein or TRPC6 a membrane bound channel mean?

The interaction of PON2 with TRPC6 could protect the lipid environment of TRPC6 from peroxidation and thus prevent TRPC6 hyperactivity in states of oxidative stress. Mechanistic evidence for this hypothesis is provided in the second manuscript submitted to this special issue (“Capsazepine (CPZ) inhibits TRPC6 conductance and is protective in adriamycin-induced nephropathy and diabetic glomerulopathy.”). The functional interaction of PON2 and nephrin is an interesting question for future research.

6) Is calcium signaling altered in PON2 deficient podocytes?

We did not investigate downstream targets of TRPC6 signaling. We feel this is beyond the scope of this manuscript.

Reviewer 2 Report

The manuscript by Hagmann and co-authors decribes Paraoxonase 2 deficiency as new therapeutic target in podocytopathies. The paper is very diligently written, the Figures, including suppl. figures, guide the reader through the manuscript and help the reader to understand the content.

I have a few comments to further improve the paper:

1) English/style: the manuscript would benefit from editing. For example, in the abstract, line 23, "largely obsure" and line 24 "cells called podocytes";  Line 335 "nearly an order of magnitude greater" (sounds like google translater). Each text is correct, but can be improved. In the results section, you change from past to present. 

2) Material & Methods: lines 129, 238, 239: you write, where the murine podocytes and knockout mice originate from, however, you might explain further (gift by ...., or matterial transfer agreement MTA in place).

3) Figures:

a) please include the numbers in figure 1 and explain further: is this finding a description of one slide or have severeal slides/biopsies?  Where do the kidney biopsy samples come from? Patient consent/IRB needed?

b) Include numbers in figure 2, 5D, suppl. Fig S1A, S2B, S4 to allow the reader to understand the statistics.

4) Discussion: please include a paragraph about the "from bench to bedside" translational medicine: how can your research benefit patients with FSGS? For example, there are already phase 2 clinical trials targeting the podocyte lipid pathway, such as R3R01-ASFSGS-201, NCT05267262 in patients with FSGS and Alport, titled "A Phase II, Multi-center, Open-Label Study to Assess Safety, Tolerability, Efficacy and Pharmacokinetics of R3R01 in Alport Syndrome Patients with Uncontrolled Proteinuria on ACE/ARB Inhibition and in Patients with Primary Steroid-Resistant Focal Segmental Glomerulosclerosis" Why not discussing your findings in the context of this ongoing trial?? 

5) Additional information at the end of manuscript: Please check, if you need to include the funding source, acknowledgements: for example for the gifts such as knockout animals and cell line, IRB-votes for human samples, animal authorities (already in the M&M text).

minor: misspelling in line 331: "fullly"

Author Response

We thank the reviewers for their thoughtful comments. We provide a point-by-point response with our reply in italics.

Best regards,

for all authors-

Henning Hagmann

The manuscript by Hagmann and co-authors decribes Paraoxonase 2 deficiency as new therapeutic target in podocytopathies. The paper is very diligently written, the Figures, including suppl. figures, guide the reader through the manuscript and help the reader to understand the content.

I have a few comments to further improve the paper:

1) English/style: the manuscript would benefit from editing. For example, in the abstract, line 23, "largely obsure" and line 24 "cells called podocytes";  Line 335 "nearly an order of magnitude greater" (sounds like google translater). Each text is correct, but can be improved. In the results section, you change from past to present. 

We thank the reviewer for this helpful critique. We have edited the manuscript for style and orthography.

2) Material & Methods: lines 129, 238, 239: you write, where the murine podocytes and knockout mice originate from, however, you might explain further (gift by ...., or matterial transfer agreement MTA in place).

We have specified the text accordingly.

3) Figures:

  1. a) please include the numbers in figure 1 and explain further: is this finding a description of one slide or have severeal slides/biopsies?  Where do the kidney biopsy samples come from? Patient consent/IRB needed?

We thank the reviewer for this insightful comment. We have quantified PON2-positive cells per glomerular area in several human biopsy samples and provide statistical analysis in novel Suppl. Fig. 1. Origin of kidney biopsy samples and IRB/patient consent are stated in M&M.

  1. b) Include numbers in figure 2, 5D, suppl. Fig S1A, S2B, S4 to allow the reader to understand the statistics.

We have now provided these numbers in the figure legend.

4) Discussion: please include a paragraph about the "from bench to bedside" translational medicine: how can your research benefit patients with FSGS? For example, there are already phase 2 clinical trials targeting the podocyte lipid pathway, such as R3R01-ASFSGS-201, NCT05267262 in patients with FSGS and Alport, titled "A Phase II, Multi-center, Open-Label Study to Assess Safety, Tolerability, Efficacy and Pharmacokinetics of R3R01 in Alport Syndrome Patients with Uncontrolled Proteinuria on ACE/ARB Inhibition and in Patients with Primary Steroid-Resistant Focal Segmental Glomerulosclerosis" Why not discussing your findings in the context of this ongoing trial?? 

We thank the reviewer for providing this important clue. We have included the following paragraph in the discussion:

There is still a lack of therapeutic options for most glomerular diseases. In recent years, alterations of the lipid metabolism in glomerular cells have gained attention. Currently, the first clinical phase II trials targeting altered lipid metabolism in glomerular diseases are being conducted. The lipid-modifying small molecule R3R01 affects cellular lipid content and is tested in patients with steroid resistant FSGS and uncontrolled proteinuria in Alport syndrome (NCT05267262). However, the pathophysiology referring podocyte disease in states of altered lipid metabolism are not yet understood. Our data suggest a pathophysiologic mechanism involving increased TRPC6 activity in response to altered lipid content and substantiate the significance of research directed to lipid metabolism in glomerular disease to open novel therapeutic avenues.

5) Additional information at the end of manuscript: Please check, if you need to include the funding source, acknowledgements: for example for the gifts such as knockout animals and cell line, IRB-votes for human samples, animal authorities (already in the M&M text).

Additional information was amended after the discussion section.

minor: misspelling in line 331: "fullly"

Thank you, the typo was corrected.

Round 2

Reviewer 1 Report

The authors have responded to my comments. The only additional comment that I have is that the foot process effacement in the PON2 KO mice is very mild if present at all but I do not feel that this is a major issue.